# Promoting Equity and Assuring Teaching and Learning Quality: Magisterial Lectures in a Philippine University during the COVID-19 Pandemic

Maria Mercedes T. Rodrigo [1,*] and Estelle Marie Macuja Ladrido [2]

1  Department of Information Systems and Computer Science, Ateneo de Manila University, Quezon City 1108, Philippines
2  Department of Communication and Areté, Ateneo de Manila University, Quezon City 1108, Philippines; eladrido@ateneo.edu
*  Correspondence: mrodrigo@ateneo.edu

**Abstract:** When the COVID-19 pandemic forced universities to shift to online learning, one of the challenges to faculty and administrators was to provide students with high-quality, curriculum-based learning materials that could be accessed despite students' variable levels of Internet access. Part of the Ateneo de Manila University's response to this challenge is the production of the Magisterial Lectures, an Open Educational Resource (OER) series of video lectures by some of the University's most respected faculty members. The goals of this paper are to describe how the production of the lectures was guided by the principles of quality and equity; to discuss the use and reach of the lectures based on YouTube analytics and a survey of Ateneo students and teachers; and to measure the impact of the lectures on students' learning experience. We enact quality in terms of curricular alignment and high production value. Equity was achieved by making the resource available publicly, free of charge. We found that the videos reached over 350,000 viewers in 37 countries. A survey of Ateneo students and teachers, the primary beneficiaries, shows that these materials were effective educational tools. Their effectiveness is attributable to the grounding of the production in quality and equity; the teachers' careful integration of the recordings in their lessons; and the students' engagement with the lectures following their own learning preferences and strategies.

**Keywords:** COVID-19; magisterial lectures; open educational resources; video-based learning

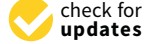

## 1. Introduction

UNESCO defines Open Educational Resources (OERs) as "learning, teaching and research materials in any format and medium that reside in the public domain or are under copyright that have been released under an open license, that permit no-cost access, re-use, re-purpose, adaptation and redistribution by others" [1]. While these materials may exist in any medium, UNESCO recognizes that information and communication technologies (ICTs) play a key role in providing effective, equitable, and inclusive access to OERs and in supporting their use, adaptation, and redistribution [1]. OERs created for or with ICTs and distributed using ICTs can theoretically be accessed anytime, anywhere, and by anyone. They can motivate content creation and pedagogical innovations and promote greater educational equality. For learners, OERs provide the ability to study anytime, anywhere, at little to no cost. OERs broaden or deepen students' understanding of a domain. For educators, OERs can become part of an existing learning module or be adapted and included in new learning modules. Careful curation of OERs can reduce the teacher preparation time, freeing teachers to concentrate on improving overall learning experiences [2]. Use of OERs can also promote a motivating pedagogical approach known as resource-based learning in which students "are provided with varied and multiple resources that engage them in the learning process." [3]. Recognizing OERs' benefits, the

UNESCO General Conference held in November 2019 issued five calls to action: to build the capacity of stakeholders to create, access, re-use, adapt, and redistribute OERs; to develop supportive policies for OERs; to encourage inclusive, equitable, and quality OERs; to nurture the creation of sustainability models for OERs; and to promote and reinforce international cooperation in OERs.

While the use of OERs is already established in certain contexts such as distance education and in countries such as Sweden [4], the COVID-19 pandemic heightened both the awareness of and need for OERs. The pandemic lockdowns subjected many families to economic hardship. Costly textbooks, which some students could barely afford prior to the pandemic, slipped further out of reach. The inability to obtain these learning resources had an impact on student achievement: under-resourced students were likely to achieve poorer academic outcomes [5]. OERs offer these learners an alternative.

It would be erroneous to assume, though, that all OERs are created equal. OERs that are of an acceptable quality and promote equity needed to be designed and distributed with these goals in mind. In this paper, we discuss the Magisterial Lectures, an OER series produced by the Ateneo de Manila University in the Philippines through its innovation and creativity hub, Areté. We describe the production process, general usage, and feedback from the teachers and students who used them. This paper discusses:

(1) The production design decisions and process to illustrate the emphasis on quality and equity;
(2) The use and reach of the Magisterial Lectures as captured by YouTube analytics;
(3) The use of Magisterial Lectures by Ateneo teachers and students; and
(4) The impact of the Magisterial Lectures on Ateneo students' learning experiences.

In describing the design, implementation, and assessment of the Magisterial Lectures, this paper contributes processes and outcomes to the literature. By processes, we refer to the ways in which we enacted the principles of quality and equity and a method for assessing OER impact. By outcomes, we refer to the lectures themselves, which are now available to teachers and learners worldwide, and evidence of their ability to support instruction.

## 2. Context

The OER discussed in this paper was created by the Ateneo de Manila University, a university in the Philippines, during the COVID-19 pandemic. The primary beneficiaries of these materials were Ateneo students and teachers, most of whom were in the Philippines, working or studying from home.

### 2.1. The Philippines

The Philippines is a developing nation in Southeast Asia with a population of 110 million as of July 2021 [6]. The country is an archipelago divided into three main island groups: Luzon, Visayas, and Mindanao. These are further divided into 81 provinces grouped into 17 regions [7]. The National Capital Region (NCR), also known as Metro Manila, is located in Luzon and is the seat of national government. It is home to one-eighth of the population, approximately fourteen million people [6].

### 2.2. Internet Access in the Philippines

According to Statista [8], 63% of adults nationwide have Internet access, but the percentages per region vary. At 84%, the NCR has the highest percentage of adults with Internet access. A total of 65% of adults in Luzon in its entirety and 62% of adults in the Visayas have connectivity. Mindanao has the lowest level of connectivity, with only 47% of adults able to access the Internet.

Beyond these estimates of Internet penetration, the quality of Internet access is a separate issue entirely. At an average of 32 Mbps, Philippine Internet speeds are among the slowest in Southeast Asia [9,10]. Thailand and Singapore averaged 220 Mbps and 247 Mbps, respectively. Furthermore, Philippine Internet costs are among the most expensive in

the world. Fixed broadband subscriptions in the Philippines cost about the same as subscriptions in Thailand and Singapore.

### 2.3. Ateneo de Manila University

The Ateneo de Manila University is a private, Filipino, Catholic, Jesuit university located in Quezon City, Metro Manila. Established in 1859, the Ateneo began as a public primary school for the children of Spanish residents [11]. It eventually grew into an elite institution, placing 124th in the 2022 edition of the QS Asia University rankings [12].

Ateneo de Manila is composed of three academic units: Basic Education, the Loyola Schools, and the Ateneo Professional Schools. The Basic Education unit is composed of the Grade School, the Junior High School, and the Senior High School. The Professional Schools constitute the post-graduate unit, which is composed of the Ateneo Graduate School of Business, the Ateneo School of Medicine and Public Health, the Ateneo Law School, and the Ateneo School of Government. The Loyola Schools constitute the tertiary-level unit of the Ateneo, offering undergraduate and graduate degree programs in arts and sciences. It is composed of the School of Humanities, the School of Social Sciences, the School of Science and Engineering, the John Gokongwei School of Management, and the Gokongwei Brothers School of Education and Learning Design.

As mentioned earlier, Areté is the creativity and innovation hub of Ateneo de Manila. Serving all the academic units and the general public, Areté is a physical space composed of performing arts venues, galleries, studios, laboratories, learning spaces, and meeting areas. Areté is also a supportive community that hosts and connects diverse academics, artists, professionals, and leaders in generative exchanges that spur new ideas [13].

### 3. Review of Related Literature

In response to the COVID-19 pandemic, governments imposed strict, sweeping mitigation efforts including travel bans, social distancing, and quarantines. Schools and universities were forced to close campuses and deploy distance learning solutions in order to reach the approximately 1.5 billion students affected worldwide [14]. This sudden shift brought with it a myriad of challenges: keeping personnel safe and healthy, ensuring academic rigor and engagement, providing clear communication, and others [15].

This shift to distance or online modes involved the use of fully remote teaching solutions such as mobile learning, radio, Zoom-based lectures, or any other methods that are contextually feasible. Institutions of varying types and sizes all over the world were forced to find and implement quick solutions [16]. The crisis required the academic manpower to redesign in a short period of time what was supposed to be an already planned out academic term.

Fortunately, most faculty members were not starting from zero. A multinational, multi-institutional study of eight colleges and universities from different continents showed that faculty had some familiarity with enabling technologies such as Learning Management Systems (LMSs). Most faculty used LMSs even prior to the pandemic [17], habitually posting slides or sharing links to online resources. Faculty were able to use the technology to engage their students in collaborative learning, student presentations, and class discussions.

Educational institutions worldwide made extensive use of OERs to support both teachers and learners during the pandemic [4]. The University of South Africa made use of OERs for in-service teacher training on online learning. China opened 24,000 online courses to university students, providing 12 undergraduate degrees and 18 vocational courses. In addition, national TV channels were used to distribute learning materials to rural areas. In Turkey, broadcast technologies were the main carrier of educational content. Educational TV programs provided educational resources to K12 students so that they could continue with their formal education.

Educational institutions also redoubled their commitment to developing and sharing OERs [4]. In Brazil, for example, teachers coordinated a project to develop an OER for

academic improvement. Canadian institutions curated, developed, and shared digital learning materials for first-year undergraduate students.

The Philippines was no exception to these developments. Face-to-face classes were suspended following the proclamation placing the entirety of Luzon under enhanced community quarantine starting 15 March 2020 [18]. The Philippines Commission on Higher Education (CHED) advised higher education institutions to implement distance learning methods in an effort to provide students with academic continuity despite the suspensions.

*3.1. Use of Video-Based Lectures during COVID-19*

Even prior to the pandemic, video was the most popular form of OER [19]. They were and continue to be central to the online learning experience: students enrolled in Massively Open Online Courses (MOOCs) spend the majority of their time watching videos [20]. Prior work has shown that student satisfaction with video lectures had a strong positive relationship with learning and the learning experience [21]. Video lectures not only provided content. They also contributed to teaching presence and social presence, enabling faculty to project themselves socially and emotionally as real people.

During COVID-19, face-to-face interactions were, by and large, replaced with online lectures hosted on platforms such as Zoom or Google Meet [22,23]. Whereas the use of video-based materials was optional prior to the pandemic, COVID-19 left educational institutions, teachers, and students with no choice. This sudden, forced transition gave rise to studies regarding the use of video technologies to fulfill educational needs and the technology and individual characteristics that mediate the use and effectiveness of videos for learning [4].

*3.2. Quality: The Impact of Educational Video Quality*

What characterizes good-quality OER videos? Beyond factual correctness, video quality can be characterized in terms of content and form.

Content. OERs in general should be aligned with and relevant to the local curriculum and the local context [2]. This means they should preferably be in the local language and should contain local content. This representation is important because it minimizes the risk of training students for external job markets at the expense of local or regional demands. OERs also need to be platforms for diversity. They need to present accurate, broad surveys of knowledge from groups that might otherwise not be heard [5].

Form. In terms of form, video production can vary greatly [24,25]. Professionally recorded lectures are usually focused on main topics, require prior design, and are branded with institutional identifiers. They are meant to be substitutes for live lectures. Instructor-recorded lectures, on the other hand, tend to have a more amateurish feel to them—the video may be too bright, the sound quality might be low, and so on.

There is continuing research on the impact of video quality on student learning. The social and sensory richness of the learning environment is positively associated with student enjoyment and performance in the traditional classroom. In online settings, production value acts as a surrogate for social richness [21,26]. Wu and colleagues [25], for example, found that professionally produced videos led to higher knowledge retention and that these effects were most notable among graduate students.

During the pandemic, the studies and their findings became more nuanced. Rickley and Kemp [27] found that videos that followed principles of multimedia learning (e.g., that removed unneeded media elements to reduce extraneous cognitive load and added cues to essential matter) had a positive impact on students' perceived learning and satisfaction. The Harvard School of Dental Medicine found that synchronous live lectures increased burnout and decreased retention and engagement [28]. It was therefore better to record lectures for students to review later.

### 3.3. Equity: Challenges of Access and Inclusion

Education and training are said to be equitable when they are accessible to and inclusive of all, regardless of gender, ethnicity, geographic location, and other factors [29]. When OER videos are uploaded to YouTube or some similar public platform, they are assumed to provide maximal equity because they are universally accessible. However, this is not actually the case.

Access to technology remains one of the impediments to students in developing countries. In the Philippines, as in most of the developing world, the mobile phone is the primary computing platform [30]. While mobile phone penetration continues to rise, Internet access still varies greatly among phone owners [31]. People from lower socio-economic classes who constitute the majority of the population make use of pre-paid mobile subscriptions [32]. To access the Internet, these subscribers purchase bandwidth in small, low-cost increments, e.g., 20.00 PHP for 15 MB valid for one day. Hence, they conserve their Internet time and only turn on data when necessary.

For OERs to have the widest reach, they need to be available in multiple formats. While video represents an ideal distribution medium, OER content also needs to come in other forms that are not as bandwidth-intensive, when possible.

Another aspect of equity that is not often discussed is the monitoring of OER usage. The literature on OER use, particularly in the developing world, is relatively thin [19]. While the number of OER projects is growing, there is no comprehensive documentation of OER producers and users [33]. Data on use and reuse of OERs are usually not visible in public repositories or outside the institutions that own the OER [34]. Studies of usage would help measure the impact of these resources and may inform institutional decisions about continued OER production.

### 3.4. Synthesis

This review of prior work makes clear that there is a need for localized, contextualized, curriculum-based OERs. Video format is preferred but content should also be made available in other media, e.g., text and audio, in order to increase accessibility. There is also a need to measure the usage of these resources in order to determine their impact.

This paper hopes to contribute to the literature by describing the process of design, the implementation, and the methods of impact assessment of the Ateneo's Magisterial Lectures, produced by Areté. It also hopes to make this resource more visible for other teachers and learners to use. Finally, it hopes to contribute to what is known about how OERs are used and how they affect the learning experience.

## 4. Magisterial Lectures

When the COVID-19 pandemic forced universities to shift to online learning, one of the challenges to faculty and administrators was to provide students with high-quality, curriculum-based learning materials that could be accessed despite students' variable levels of Internet access. Part of the Ateneo de Manila University's response to this challenge was the Magisterial Lectures series. Produced by Areté in partnership with the Loyola Schools Department of Communication and the Eugenio Lopez Jr. Center for Multimedia Communication, the Magisterial Lectures are professional-quality video recordings of university lectures delivered by some of the university's most respected faculty members.

The lecture series' primary purpose was to create an OER that preserved the quality of Ateneo instruction amidst the fears and uncertainties relating to the shift to online learning at the onset of the pandemic. The lectures were intended for class use during, and even after, the pandemic as, under the Loyola Schools curriculum, many classes shared lecture topics. Instead of having each teacher record their own lectures, Areté thought it would be beneficial for some of these common topics to be recorded for sharing across the different classes and sections.

Equity and quality guided the design and implementation of the production process. Two of the equity considerations were public access and media format. Early in the project's

development, the production team discussed whether to limit access to the Magisterial Lectures to the Ateneo community only. Upon the request of sister schools outside of Metro Manila, though, the production team decided to make the materials public. All lecturers were made aware that the videos would be publicly accessible.

Regarding media format, the production team hoped to make the lectures accessible in the three forms—video, audio-only, and lecturer's script and presentation deck in pdf format—to make the contents accessible even to those with limited Internet bandwidth. Unfortunately, as of the time of this writing, we have not yet been able to upload the audio-only versions. With regard to the transcripts, some of the lectures were in Filipino or were bilingual, and the production team did not have the manpower to transcribe these lectures. YouTube was able to produce reasonably good transcripts and closed captioning for lectures that were in English. We did include the lecturer's slide decks whenever possible. Students could download these slides and refer to them while watching the lecture.

Quality was ensured in a number of ways. It began with the recruitment and preparation of the lecturers. The production team first reached out to school deans and department heads not only to source potential lecturers but also to determine the most necessary topics for the OER. From responses made by deans and department heads, invitations were issued to potential lecturers. Once a lecturer agreed to the recording, the production team coordinated the date of the recording and the use of the venue against the availability of the lecturer. The lecturer was also provided with a comprehensive briefing packet that explained the recording process and contained production notes, tips for delivering an engaging recorded lecture, presentation templates, and other guidelines to aid them with their preparations for the recording day. One important note given to lecturers as part of their preparations was to deliver their lectures extemporaneously as much as possible, simulating a classroom delivery as much as possible. The lecture director also instructed lecturers to remember that, despite the cameras, they were still talking to one student who would be accessing the lecture recording through a computer or mobile device.

Content was actualized through the selection of lecturers and their respective topics. Because the lectures were selected from among those that faculty would normally deliver face-to-face, each Magisterial Lecture was aligned with university curricula. Further, faculty members were given the option to speak in the local language if they wished.

As of the time of this writing, there are 66 lectures in the series, some of which were released in multiple parts, for a total of 76 video releases in all (see https://arete.ateneo. edu/connect/magisterial-lectures (accessed on 12 February 2022)). The majority of the videos (29 of 66) were delivered by faculty from the School of Social Sciences (e.g., *The Political Psychology of Active Nonviolence* by Dr. Cristina J. Montiel; *Finding the Prehispanic Filipino Woman: Clues from Spanish Documents* by Dr. Olivia Anne M. Habana). A total of 25 out of 66 were from the faculty of the School of Humanities (e.g., *On The Teaching of Poetry* by Dr. Rica Bolipata-Santos; *Theater, Trauma, and the Rehearsal to Recovery* by Ms. Missy Maramara; *Ethics of Buddhism* by Dr. Manuel Dy). The remaining videos were from the School of Science and Engineering (e.g., *Weaving Mathematics* by Dr. Ma. Louise Antonette N. De Las Peñas), the School of Government (e.g., *The Climate Emergency: A Social Justice and Human Rights Approach* by Atty. Antonio G. M. La Viña), and the John Gokongwei School of Management (e.g., *Unlocking the Power of the Franchising Format* by Mr. Rodolfo P. Ang).

On form, the production team was adamant that the videos should have a high production value so that the videos would age well. Hence, the lecture recordings were subject to professional standards as each was taped live using a three-camera setup in the audio-controlled environment of a theater soundstage, using a deliberate lighting design employing institutional signifiers (see Figure 1). The production setup incorporated a screen from which lecturers could view their prepared slide presentations. Further, audio and video channels were recorded separately. Apart from production management and coordination, the production team included a line producer, a lecture director, a technical director, a director for photography, camera operators, sound men, gaffers, and grips.

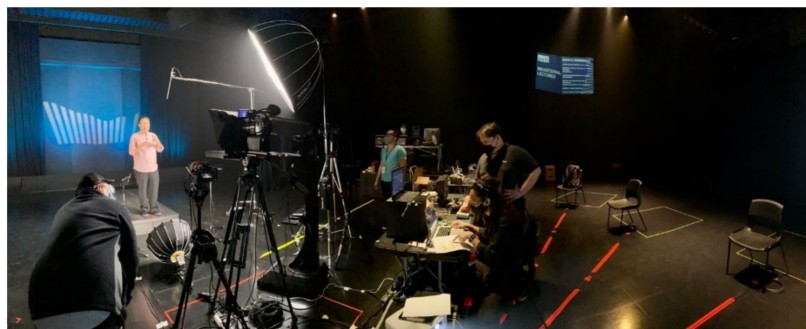

**Figure 1.** Recording environment for Magisterial Lectures.

Productions were scheduled once a week, with three lectures recorded on each production day. The productions began at 8 a.m., when the technical crew set up cameras, microphones, and lighting equipment addressing the specificities pertaining to equity and quality, and ended at 4:30 p.m. In between each lecture, the sound stage was sanitized in keeping with safety protocols. Notable here was that four members of the production team were trained safety officers, which helped ensure high safety standards on site.

The lectures were subject to post-production editing, where presentation slide decks were incorporated into the live recordings. The postproduction process typically took 3–4 weeks. Presentation slides were treated as visual aids and inserted at key points to assist viewers appreciate the lecturer's points. The lecture director however maintained directorial control over the use of supplementary video material. Once ready, executive producers and line producers reviewed each lecture carefully before permitting its release via Areté's website and YouTube channel. Each Magisterial Lecture has a dedicated page on the Connect page of the Areté website where slide decks and other related materials are available for download.

## 5. Methodology

To measure the impact of the Magisterial Lectures, we made use of data from three sources: access statistics from YouTube, a student survey, and a teacher survey. Participating students and teachers completed a self-report survey that asked about their use of the Magisterial Lectures and their thoughts and feelings about the medium and its effectiveness. Data were anonymous. Respondents were free to decline to participate, without consequence, at any time. All data collection and analysis methods were submitted to the University Research Ethics Committee. The Committee classified the protocol as exempt from a full ethics review.

### 5.1. Respondents

We employed a purposive sampling method in recruiting student and faculty respondents. Our recruitment process considered how the Magisterial Lecture producers actualized content in the Magisterial Lectures. We noted that Philosophy and Science and Society faculty members were the most enthusiastic about integrating the Magisterial Lectures in their courses. They were the first to identify the lecturers and lecture topics that they wanted recorded as an OER. Furthermore, we limited our survey to these two subject areas because undergraduate students are required to take Science and Society and certain Philosophy classes regardless of their specializations.

To be eligible to participate in the survey, faculty had to require their students to watch at least one Magisterial Lecture as part of their course. Faculty who responded to our request by saying that watching Magisterial Lectures was optional or that they did not require Magisterial Lectures at all were excused from participation. Students, on the other hand, had to be members of classes where faculty members required the viewing of Magisterial Lectures as part of their course content.

We initiated our purposive recruitment by requesting permission and soliciting support from the Chair of the Department of Philosophy and the Coordinator for Science and Society. Upon receiving their approval, we wrote the faculty members individually to ask them to respond to the faculty survey and distribute the student survey form to their students. The survey links were attached to the email invitation issued to each teacher. Both faculty and students were given three weeks to respond to the survey, but this was eventually extended to six weeks. We also initiated follow-up communications to faculty members to remind them to answer the survey and encourage their students to respond.

*5.2. Instruments*

Both faculty and students were asked to complete an adapted version of the Effectiveness of Video Lectures Instrument of [35]. Respondents had to indicate their levels of agreement with the following statements (Strongly Disagree to Strongly Agree):

a.   Quality of Video Presentation

    i    The visual design was satisfying;

    ii   The lecturer presented well;

    iii  The pacing of the lecture was appropriate;

    iv  The lecture experience was pleasing; and

    v   The production quality of the lecture made it appear more credible.

b.   Depth and Quality of the Learning Content

    i.   The content of the lecture helped me with my course;

    ii.  The flow of the content was logical;

    iii. The content was informative;

    iv. The content was relevant to my course; and

    v.  I believe (my students/I) felt motivated to learn from the lecture.

c.   Aid to Retention of Learning

    i.   The lecture aided with (student/my) retention of course concepts; and

    ii.  The lecture enhanced (student/my) knowledge and skills.

d.   Ease of Use and System Support

    i.   The lecture was easily accessible; and

    ii.  (Student/my) computers supported the lecture files seamlessly.

We collected some demographic information from both faculty and students, including the city/region/country from which they were accessing the Lectures and the type of Internet connection they were using. We asked faculty for the number of years that they had been teaching. We asked students for their year levels and under which of the Ateneo's five schools their majors were. In addition, the faculty were asked which Magisterial Lectures they required. They were also asked to describe the curricular context of the Lectures, i.e., for what lesson the Lecture was, what learning activities preceded the Lecture, and what assessments succeeded the Lecture.

The survey was available through a link connecting respondents to a Google Form included in the email communications to teachers who met the selection criteria.

## 6. Results

The analysis of YouTube statistics and the results from the faculty and student surveys provide some evidence of the breadth and depth of the Magisterial Lectures' reach.

*6.1. YouTube Statistics*

We referred to the statistics automatically generated by YouTube in order to measure the lectures' reach and found that the lectures had a global audience. Between July 2020, when the lectures were first released, and 14 January 2022, there were 373,699 lecture viewers. The large majority of viewers (88.7% or 331,480) were from the Philippines. A total of 1.7% (6510) were from the United States. Analytics show that the lectures were

viewed from a list of 37 other countries, with the most viewers coming from Singapore (575), Canada (571), the United Kingdom (342), Indonesia (300), India (249), Australia (210), and Japan (184). A total of 53.4% of the viewers were university-aged at 18–24 years old, while 29.8% of viewers were between the ages of 25–34 years old. More mature viewers aged 45–54 years old made up 8.4% of the viewership. In terms of gender, 45.4% of the viewers were female, while 54.6% were male.

Laptop computers and mobile phones were the devices most used to view the lectures, 45.9% and 45.3%, respectively. Tablets (4.5%) and Smart TVs (2.8%) were least used for viewing.

The lectures have gathered a total of 3,335,306 impressions, with a 3.5% click-through rate. To date, the following lectures have been the most viewed:

(1) *Tao Po! Tuloy! Halina sa Loob* by Fr. Albert Alejo, SJ (43,951 views);
(2) *Tools for Ethical Decision Making* by Dr. Antonette Palma-Angeles (33,986 views);
(3) *Rizal Without the Overcoat* by Dr Ambeth Ocampo (21,804 views);
(4) *Liberating Ourselves from the Past* by Dr. Ambeth Ocampo (19,769 views); and
(5) *Doing Philosophy* by Dr. Antonette Palma-Angeles (14,316 views).

### *6.2. Student Responses*

The researchers depended on the faculty members to invite their students to participate in the survey. Hence, there was no direct contact between the researchers and the students. For school year 2020–2021, the Loyola Schools' student population was 9152 students. However, the survey only reached a portion of this total and the researchers do not know exactly how many students were invited. Of those invited, 225 participated in the survey.

The students' ages ranged from 18 to 48, with an average of 21 years. Ninety-nine students (43.8%) were in their second year of college. Fifty-seven or 25.2% were in their third year, while 47 or 20.8% were in their fourth year. Eleven respondents or 4.9% were graduate students.

The majority of students (168 or 74.3%) had high-speed Internet access (see Figure 2) and many students had more than one means of connecting online (see Figure 3).

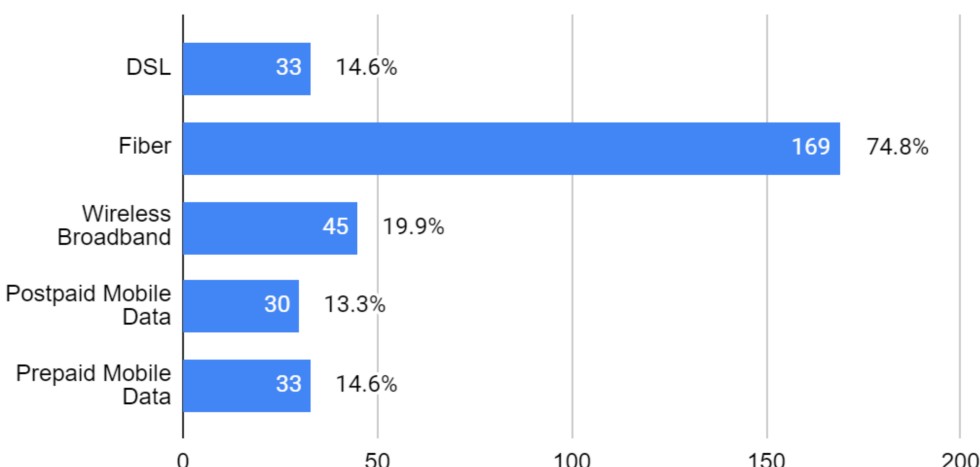

**Figure 2.** Types of Internet connectivity of the students.

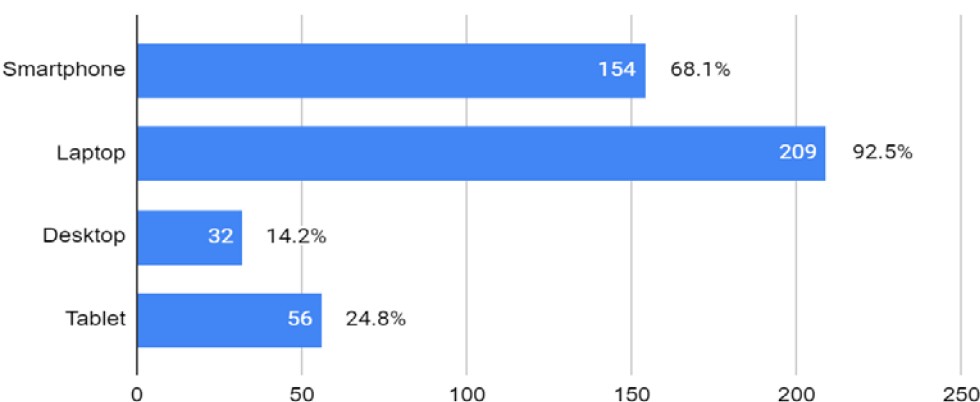

**Figure 3.** Types of computers of the students.

Most students (209 or 92.5%) made use of laptop computers but also accessed their courses using their smartphones, desktops, and tablets (see Figure 3).

A total of 148 students or 65% were based in Metro Manila. Seventy-one students or 31% were outside of Metro Manila but still within the Philippines. Nine students or 4% were outside of the Philippines. None of the students reported having difficulty accessing the materials, though one of the students did comment that "text-only versions would be helpful to those with slow internet or who rely on mobile data."

Table 1 presents the mean scores of student responses to the survey. We assigned values to responses as follows: Strongly Agree (5), Agree (4), Neutral (3), Disagree (2), and Strongly Disagree (1).

**Table 1.** Mean scores of student responses.

|  | Question | Mean Score | Standard Deviation |
|---|---|---|---|
| Video Quality | Visual design was satisfying | 4.5 | 0.63 |
|  | Lecturer presented well | 4.75 | 0.48 |
|  | Pacing of lecture was appropriate | 4.64 | 0.57 |
|  | Lecture experience was pleasing | 4.62 | 0.58 |
|  | Production quality made it feel more credible | 4.74 | 0.47 |
| Depth and Quality of Learning Content | Content helped me with my course | 4.61 | 0.58 |
|  | Flow was logical | 4.74 | 0.48 |
|  | Content was informative | 4.76 | 0.48 |
|  | Content was relevant to my course | 4.76 | 0.54 |
|  | I felt motivated to learn from the lecturer | 4.5 | 0.71 |
| Aid to Retention | Aided in retention of course concepts | 4.42 | 0.70 |
|  | Enhanced my knowledge and skills | 4.49 | 0.69 |
| Ease of Use | Lecture was easily accessible | 4.76 | 0.48 |
|  | My computer system supported lecture files seamlessly | 4.73 | 0.53 |
|  | Listening to the lecture served a purpose | 4.67 | 0.60 |

The scores show that students viewed the Magisterial Lectures as helpful OERs. They considered the lectures as informative, relevant to programs of study, as well as assistive in learning course content. Our cross tabulations between items pertaining to content and knowledge retention and skill enhancement indicate that students agreed that the integration of production quality and content in the lecture recordings helped them learn the course material. Table 2 presents an example, conveying how students agreed that the lecture presentation assisted in their retention of course concepts. This was a repeating pattern across the data set and can be explained by the standard deviation.

**Table 2.** Cross tabulation presenting video quality and retention of concepts.

| | | The Lecturer Presented Well | | | | |
|---|---|---|---|---|---|---|
| | | Disagree | Neutral | Agree | Strongly Agree | Total |
| The lecture aided my retention of course concepts | Disagree | | 1 | | 1 | 2 |
| | Neutral | | 3 | 6 | 13 | 22 |
| | Agree | | 1 | 34 | 46 | 81 |
| | Strongly Agree | | | 6 | 114 | 120 |
| | Total | | 5 | 36 | 174 | 225 |

The qualitative responses supported our assertion that high production quality enhances OER credibility, facilitates engagement, and promotes learning. Students wrote that they perceived top-notch quality and surmised a high production budget. As one student put it, "I commend the presentation of the lecture. In terms of cinematography, it made use of different angles which I applaud. I also like the clarity of the audio and video, making it pleasing audio visually. The depth of the lecture and the knowledge of the lecturer as well was very moving.".

Students also provided constructive feedback: to include more presentation slides as visual aids, include subtitles, and improve the sound quality.

Significantly, students said that the lectures provided an experience of an onsite class. This emerged from student responses such as "the lecture recording simulated a real lecture instead of a Zoom call class. It made me feel like I was learning onsite with a professor who was really passionate about their subject" and "[the lecture recordings are] a great way to give others a glimpse of what usually takes place within the classrooms in Ateneo. The lectures are thought-provoking and uniquely Atenean. My only feedback would be to ask for more!" We read these responses in light of the university's context, amidst its second year of conducting courses online. It was apparent that the lecture recordings simulated the experience of a classroom session, or else facilitated the imagination of one, and we considered this a promising aspect of the OER for further exploration.

In relation to equity, Table 1 presents that the students did not experience problems with accessing the lecture recordings online, as these were supported by their computers and supplementary devices. Further, our data show that most accessed the internet using the most stable and reliable connections in the Philippines, such as fiber and DSL, yet they also relied on alternative sources such as prepaid and postpaid mobile data to supplement DSL and fiber optic connections. This implied that students may have relied on mobile data when they experienced fluctuations in their DSL or fiber optic connections.

In terms of learning, what we found striking was how students adapted their use of the materials to suit their own learning preferences. One student said, "I opted to turn on the automatically generated [English] subtitles to aid my viewing even if this is sometimes inaccurate . . . ". Another student also commented about using the transcripts (we assume they were referring to the transcripts automatically generated by YouTube) because they made note-taking easier and this student said that they learned better when taking down notes. At least one other learner said that they preferred to listen to the lectures rather than watch them. This shows that students made use of alternative media formats to suit their learning needs and preferences.

*6.3. Faculty Responses*

We invited a total of 64 faculty members who taught Philosophy (41) or Science and Society (23). Of these, 33 responded, 17 from Philosophy and 16 from Science and Society. Faculty members ranged in age from 22 to 74. Their average age was 41. Teaching experience ranged from 1 to 54 years, with an average of 13. Most faculty members had

access to high-speed Internet, with some faculty members having more than one type of connectivity at their disposal (see Figure 4).

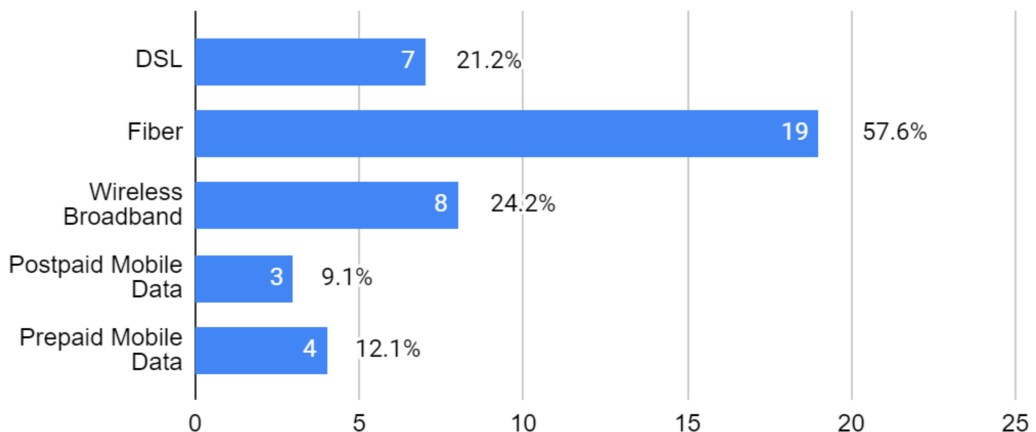

**Figure 4.** Types of Internet connectivity of the faculty.

All the faculty accessed their courses using a laptop (see Figure 5). In addition, some faculty also made use of their smartphones, desktop computers, and tablets.

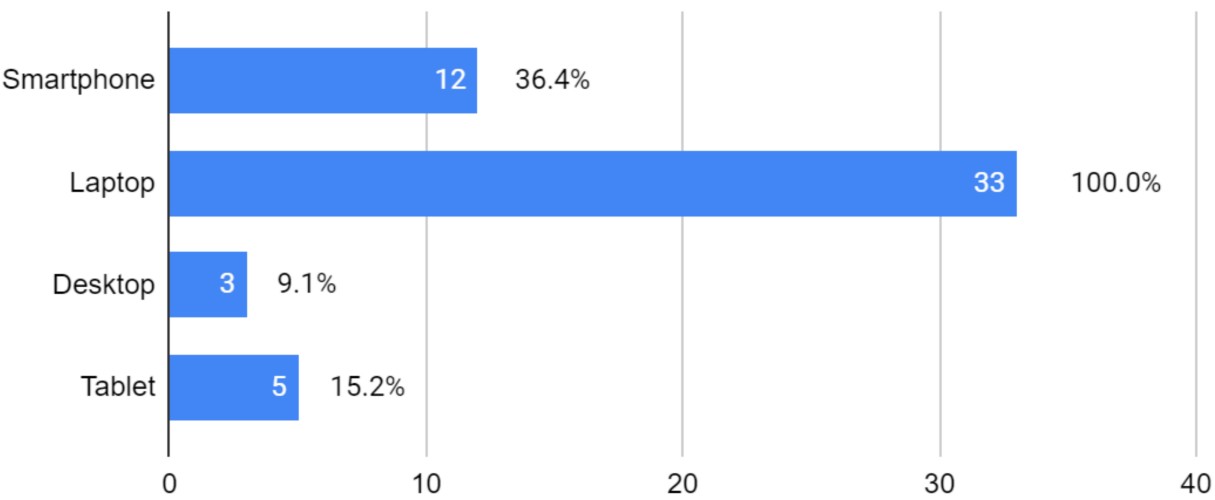

**Figure 5.** Types of computers of the faculty.

Twenty-four faculty members (73%) were based in Metro Manila while nine (27%) were outside Metro Manila but still within the Philippines. None of the faculty who responded to the survey reported having problems with accessing the materials.

Among faculty members, the consensus is that the lectures were effective tools for teaching course subject matter. However, the mean scores (Table 3) show that their responses were not as strong as students'. While teachers agreed that as an OER, the Magisterial Lectures contributed to a positive learning experience in terms of their learning content and high production quality, their responses showed more disparity, as presented by the higher standard deviation. This however may be due to the smaller sample size.

**Table 3.** Mean scores of faculty responses.

| | Question | Mean Score | Standard Deviation |
|---|---|---|---|
| Video Quality | Visual design was satisfying | 4.36 | 1.02 |
| | Lecturer presented well | 4.54 | 0.79 |
| | Pacing of lecture was appropriate | 4.52 | 0.83 |
| | Lecture experience was pleasing | 4.60 | 0.79 |
| | Production quality made it feel more credible | 4.54 | 1.06 |
| Depth and Quality of Learning Content | Content helped me with my course | 4.48 | 0.90 |
| | Flow was logical | 4.61 | 0.83 |
| | Content was Informative | 4.70 | 0.77 |
| | Content was relevant to my course | 4.73 | 0.76 |
| | I believe my students felt motivated to learn from the lecture. | 4.21 | 0.96 |
| Aid to Retention | Aided with student retention of course concepts | 4.27 | 0.91 |
| | Enhanced student knowledge and skills | 4.24 | 0.93 |
| Ease of Use | Lecture was easily accessible | 4.66 | 0.82 |
| | My computer supported the lecture files seamlessly | 4.21 | 1.02 |

Items pertaining to depth and quality rated highest among faculty respondents, which is unsurprising as the recordings were used to supplement other pedagogical tools. As one teacher reported, "[t]he Lectures were very helpful since they summarize and explain difficult technical concepts in our readings." What we found striking from teachers was their comparatively lower scores for items relating to students' motivation and retention. While they agreed that students were motivated by the lectures, and were aided by them, this agreement appeared tentative to us. Their qualitative comments though were at odds with the mean scores, as teachers' answers intimated how the lectures motivated their students to learn. They reported that students asked questions to clarify concepts, definitions, and relationships presented in the lecture recordings. One teacher wrote in his response that "[i]n my experience, even when the Magisterial Lectures were not required my students would still watch them. It was evident because they would always make references to those lectures in their papers and midterm exams". These indicated to teachers that students, apart from being required for specific courses, made use of the lecture recordings also as references for their course work in general. This supported the teachers' claim that the lectures aided in students' retention of course material and overall enhanced knowledge and skills. We surmised that this hesitance may be due to teachers' relative inexperience with teaching online, given that the study was conducted approximately one year into the university's shift to online course delivery. Additionally, as the survey was released just before mid-term exams, it was possible that teachers did not have the assurance of final grades to assess how much the Magisterial Lectures recordings contributed to student learning.

Importantly, we were struck by qualitative data revealing that, for the most part, teachers carefully prepared their students to engage the Magisterial Lectures through assigned readings and explanatory discussions during synchronous sessions. Quoting from one teacher's response, "I need to introduce the videos in a [synchronous session] to explain the value and key points that were not clear." Engagements with the lecture recordings were followed by formal and informal assessments in the form of quizzes, reflection papers, case studies, and group discussion. This reflected strong teaching presence in the use of the Magisterial Lectures as OERs; first in the area of course design (deliberate inclusion of

the lecture recordings the progression of the course), and second in preparing students to engage online resources so that they could maximize benefits.

In relation to equity, teachers also responded favorably to questions relating to ease of use, claiming that the lectures in their various formats were easily accessible for them and for their students. We were interested, though, in the relatively low score given to teachers' computer systems' ability to access the lecture files, along with a higher standard deviation. This implied that teachers themselves experienced some difficulty in accessing the lecture files using their laptop computers and may have had to rely on their smartphone to view the recordings. Relatedly, some teachers reported that the lecture recordings were difficult to find on the Areté website and were easier to locate on YouTube.

## 7. Discussion

While both teachers and students acknowledged that the Magisterial Lectures were helpful in presenting course material, what emerges from the findings is that their effectiveness as learning objects is attributable to three things. The first is in the area of production: how quality and equity were incorporated into the conceptualization of the lectures, and later executed in the production and postproduction stages. The second is in the area of course design and preparation. This pertains to the work of the teacher in presenting the lecture recordings to students. The third is on the factors that supported students' engagement with the lectures.

Quality was ensured through both content and form. Content referred to contextualization and curricular alignment. As the survey respondents were those departments who responded enthusiastically to the opportunity to integrate recorded lectures for their online course work, the findings supported how OERs, when closely integrated with curriculum and individual course content, are effective in promoting learning among students. Multiple studies make similar claims, asserting that curricular relevance decreases barriers and increases the likelihood that teachers and students will use OERs [36,37].

Quality of form referred to the high production value of the videos, something that students and teachers alike appreciated. Again, this aligned with prior studies that show the positive effects that production quality had on students' learning experience [1,21].

Certain production practices were additionally supportive of learning. One of these was the producers' insistence to keep recordings to within 20 min. Students' responses reveal that part of the appeal of the Magisterial Lectures was their "short and sweet" delivery. Reminding lecturers that they were still speaking to students listening to them from behind the cameras was an effective practice as well, as comments relating to experiencing real lectures in Ateneo classrooms convey. The positive effects of the lectures' relatively short length and their conversational tone corroborated the findings of [27] that segmentation (partitioning materials into short, learner-paced segments) and personalization (instructor's tone and style are more conversational rather than formal) of video lectures contributed to students' perceived learning and satisfaction.

While the lecture recordings were available for teachers to use for their online course work, teachers were also quite deliberate in the way these were integrated into their respective classes. As supplementary learning materials, the Magisterial Lecturers could have been made optional for students to engage. Instead, the findings show that while the lectures were deliberately included in course modules, instructors prepared students for their encounter with the lecture recordings and conducted post-viewing activities such as group discussions and formative assessments. That course design and instructors' priming and post-processing of students' engagement with the online resources are determinants of OER effectiveness is a common theme in the OER literature. Huang and colleagues [38] document stories of how teachers from various countries make effective use of OERs by providing students with a context, integrating OERs into class activities and small group interactions, and then engaging students in peer assessments and reflection.

The third point is on factors that supported student engagement with the lectures, factors that have implications for equity. To enable students to engage with an OER,

they must first have access to the OER. The lectures were available free of charge on YouTube and, based on the survey responses, students did not have difficulty accessing them. Interestingly enough, faculty gave accessibility lower scores than students did. If this is indicative of problems with connectivity, the survey findings also imply that faculty managed these difficulties by using alternative Internet connections.

Another factor contributing to equity is the faculty-created activities surrounding and scaffolding the viewing of the lectures. Creating content and making it available for free does not guarantee its use or its effectiveness. Rather, it is " . . . the judicious application of OER, in combination with appropriate pedagogical methodologies, well-designed learning objects and the diversity of learning activities, [that] can provide a broader range of innovative pedagogical options to engage both educators and learners to become more active participants in educational processes and creators of content as members of diverse and inclusive knowledge societies." [1]

Hence, the pre- and post-viewing activities were essential to supporting meaningful and effective learning.

A final factor with an equity implication is students' own sense of agency. The survey results show that students had the maturity and self-awareness to adapt the ways they used the lectures. They used YouTube-generated transcripts or else listened to rather than watched the videos to maximize play on what they knew to be their preferences. Students without the same level of self-awareness may require some coaching in order for them to maximize the educational benefits of OERs. As we move forward, we, the producers, should follow through with plans of producing transcripts and audio-only versions of the lectures, to cater to learners' individual differences.

As an OER, the experience of the Magisterial Lectures indicated that even though production quality lends to lecture credibility and aids in securing viewer engagement, student learning was maximized when there were efforts to integrate lecture content with program curricula and course content, when there was planning and deliberation on the part of instructors to incorporate OERs into classes, and when varying formats allowed students to adapt learning materials to suit their learning preferences and strategies.

## 8. Conclusions

In this paper, we described the Magisterial Lectures, an OER series of videos from the Ateneo de Manila University in the Philippines, through its creativity and innovation hub, Areté, contributing a description of the process and outcomes to the literature. In terms of process, this paper provides a concrete example of how the principles of quality and equity are woven into design decisions regarding educational materials production. The Magisterial Lectures production team ensured content quality through careful pre-production planning with academic departments, schools, and lecturers. Quality of form was achieved through the use of professional production standards and a strict post-production review process.

The paper contributes a method for assessing impact. Using an existing survey instrument and reaching out to teachers and students from the target audience through purposive sampling, we were able to collect data regarding the lectures' use and reception.

In terms of outcomes, the paper raises awareness regarding the existence of the lectures, which are now available online, free of charge. One of the first barriers to OER use is finding the OERs. Increasing the discoverability of a resource saves teachers time and effort [1,36].

The paper also provides some evidence of the use of these resources outside of the Ateneo: an analysis of the YouTube viewing statistics showed that the Magisterial Lectures reached viewers in at least 39 countries. While these numbers are not indicative of whether or how these materials are used for classes, the data address an existing gap in the literature [1,33] for documentation regarding the adaptation and use of OERs outside of home institutions.

The paper also contributes evidence of the Magisterial Lectures' impact on the primary intended audience, Ateneo teachers and students. Data about the impact of OERs in a

developing world context are still scarce [1,18]. The consensus of both types of survey respondents was that the materials' content and production quality contributed to overall learning and the learning experience, and these findings can be used to rationalize continued production of these resources.

For the benefit of classroom practice, the paper identifies three factors that contribute to the effectiveness of educational materials production and use. The first is the use of the guiding principles quality and equity during the conceptualization, production, and post-production of the lectures. Second, the effectiveness of the materials was determined in large part by the extent to which teachers integrated the materials into their lessons. A final determinant of effectiveness was the ways in which students engaged with the lectures, following their own learning preferences and strategies.

As we make our way forward, we can increase the accessibility of these materials by following through with the initial plans to produce lecture transcripts and audio-only versions. Since Areté produces its own educational podcasts, the audio-only Magisterial Lectures may be included in these channels. Furthermore, Areté should regularly monitor the use of these materials to gauge their impact and reach, and to determine whether continued production of these lectures would benefit a post-pandemic audience.

**Author Contributions:** M.M.T.R. identified the survey instrument and adapted it for this study. She reminded the faculty to respond to the survey and field it out to their students. She wrote the introduction, review of related literature, and methodology sections of the paper. She also wrote part of the results section. E.M.M.L. fielded out the surveys to the faculty. She performed the analysis of student and faculty surveys and the thematic analysis of their qualitative responses. She wrote the description of the Magisterial Lecture production and discussion sections of the paper. All authors have read and agreed to the published version of the manuscript.

**Funding:** This research received no external funding.

**Institutional Review Board Statement:** The study was conducted in accordance with the Declaration of Helsinki, and approved by the University Research Ethics Office of the Ateneo de Manila University (protocol code AdMUREC_21_056 and 6 October 2021).

**Informed Consent Statement:** Informed consent was obtained from all subjects involved in this study.

**Acknowledgments:** We thank our colleagues in the Magisterial Lectures production team: Ricardo G. Abad, Aaron R. Vicencio, Dennis Temporal, Glen Charles Lopez, D Cortezano, Ivy Baggao, Adriane Ungriano, Arielle Acosta, Vanessa Reventar, RJ Adarlo, Ian Valderama, Jeffrey Lorenzo, Alejandro Adolfo, Smile Indias, John Robert Yam, Jethro Nibaten, Justine Ray Santos, Nilo Beriarmente, Rey Sotto, Kevin Duane Ligot, and Christian Valderama. We thank the faculty and students who participated in this study. Finally, we thank the lecturers who so generously gave of their time to be a part of the Magisterial Lecture series.

**Conflicts of Interest:** The authors declare no conflict of interest.

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
