# Peer review of "Promoting Equity and Assuring Teaching and Learning Quality: Magisterial Lectures in a Philippine University during the COVID-19 Pandemic"

_education, doi:10.3390/educsci12020146_

Round 1

Reviewer 1 Report

The Magisterial Lectures series provided an opportunity to construct a new OER curriculum, state-of-the-art online lectures in epidemiological safety, and complex analyses based on it. An essential feature of the development is that, in addition to creating a public base for the university’s target populations (faculty and students involved), it has also provided significant international promotion courtesy of YouTube. A review of the relevant literature, partly due to COVID references, is relatively narrow and contingent on the reader. Lectures based on public video presentations are the content focus of the development and analysis; the related quality aspects can be considered significant. Another positive element of the development is that the authors prioritise the issue of equity, and they are trying to adapt the content service on YouTube to the actual characteristics of Internet use. Although the Areté hub plays a vital role in communication, relatively little is known about this innovative background and the content-thematic structure of Magisterial Lectures, which is otherwise well documented in terms of its technological process. The one presented, the only illlecture illustrated with photos, seems scarce. The empirical data presented are undoubtedly interesting; however, Internet and computer use characteristics do not show a significant pattern for either teachers and students or international references. The article does not explain student sampling and the number of respondents. The five-point attitude assessment shows good primary results but no more general conclusions. The two chapters concluding the study (Discussion, Conclusion) contain general summaries that highlight the essential elements of the innovation process but do not go beyond the documentation. Notwithstanding the above, a report documenting the current development process can be considered a helpful reference. I suggest filling in the gaps in the content (hub Areré, lecture series thematic network) and presenting the background of the empirical data in more detail.

Author Response

Thank you for your feedback. We have attempted to respond to each of your points. The responses are summarized in the attached document.

Reviewer 2 Report

The theme is intereanting and very actual! 

Your introduction and  literature review would be improved by the following:
• state the research contributions more clearly
• align the abstract to the contents of your article
• be precise about how the research was conducted
• identify the research gap and the goals of the research

It is must to introduce more recent bibliography. 

All the best! 

• describe knowledge acquisition behaviour concisely
• correct grammar and punctuation errors, 

Author Response

(The authors gave the same response as above.)

Reviewer 3 Report

The publication submitted to me for review analyses issues that are currently significant and relevant. I think this is a highly important and interesting paper. The article is prepared properly and clearly; therefore, I note just a few details that I think need to be clarified:

  • The introduction to the publication lacks a more thorough substantiation and presentation of the exploration level of the situation regarding the impact of the COVID19 pandemic on educational processes, especially in Master degree studies.
  • I would recommend not to present such type of illustrations in a scientific publication, they can be successfully referred to and described by providing a link where readers could get acquainted themselves.
  • It is necessary to rethink the presentation of Table 1, which visually is not properly presented, all the more so because it is not even clear what values are presented. I think it is unnecessary to provide results according to all options of answers because essential trends are shown by the mean values.
  • When presenting results in the part YouTube statistics, the criteria and steps of the analysis are not clear – they should be presented more clearly and in more detail.
  • In the part of discussion, it would be worth clearly showing what criteria were used to assess the quality of magisterial lectures. How does this relate to the results of the already conducted studies. In the discussion part, it would be good if the link with the results of other studies is presented while discussing perspectives and aspects to be improved.

Considerable work has been done, congratulations to the authors and I wish good luck.

Author Response

(The authors gave the same response as above.)

Reviewer 4 Report

A well-written paper that can add to the body of evidence of the importance of OER in the changing global landscape, be it pandemics or the way young learners now choose to learn and obtain knowledge.

Please proof-read as I found some minor grammatical errors. e.g. in the abstract: "An examination of YouTube analytics we found that the materials had global reach, as they were accessed by people all over the world."  This could be better written as "An examination of YouTube analytics demonstrate that the materials had global reach, as they were accessed by people all over the world." Please note that the abstract should be written in the present tense, while the paper is written in the past tense.

Author Response

(The authors gave the same response as above.)

Round 2

Reviewer 1 Report

The authors provided detailed and concrete answers to the suggestions initiating the corrections. The information about the Aretè hub is an essential feature, and a more detailed presentation of the student sample helps interpret the empirical work well. The other professional impact of the study and the researchers' future work is substantially supported by the supplementation of the discussion and conclusion.

Author Response

Our thanks to the reviewer.